# Location and expression kinetics of Tc24 in different life stages of *Trypanosoma cruzi*

Leroy Versteeg[1,2,3], Rakesh Adhikari[1,2], Cristina Poveda[1,2], Maria Jose Villar-Mondragon[1,2], Kathryn M. Jones[1,2], Peter J. Hotez[1,2,4], Maria Elena Bottazzi[1,2,4], Edwin Tijhaar[3], Jeroen Pollet[1,2]*

**1** Department of Pediatrics, National School of Tropical Medicine, Baylor College of Medicine, One Baylor Plaza, Houston, Texas, United States of America, **2** Texas Children's Hospital Center for Vaccine Development, Baylor College of Medicine, Houston, Texas, United States of America, **3** Cell Biology and Immunology Group, Wageningen University, Wageningen, The Netherlands, **4** Department of Biology, Baylor University, Waco, Texas, United States of America

* Jeroen.Pollet@bcm.edu

## Abstract

Tc24-C4, a modified recombinant flagellar calcium-binding protein of *Trypanosoma cruzi*, is under development as a therapeutic subunit vaccine candidate to prevent or delay progression of chronic Chagasic cardiomyopathy. When combined with Toll-like receptor agonists, Tc24-C4 immunization reduces parasitemia, parasites in cardiac tissue, and cardiac fibrosis and inflammation in animal models. To support further research on the vaccine candidate and its mechanism of action, murine monoclonal antibodies (mAbs) against Tc24-C4 were generated. Here, we report new findings made with mAb Tc24-C4/884 that detects Tc24-WT and Tc24-C4, as well as native Tc24 in *T. cruzi* on ELISA, western blots, and different imaging techniques. Surprisingly, detection of Tc24 by Tc24-C/884 in fixed *T. cruzi* trypomastigotes required permeabilization of the parasite, revealing that Tc24 is not exposed on the surface of *T. cruzi*, making a direct role of antibodies in the induced protection after Tc24-C4 immunization less likely. We further observed that after immunostaining *T. cruzi*–infected cells with mAb Tc24-C4/884, the expression of Tc24 decreases significantly when *T. cruzi* trypomastigotes enter host cells and transform into amastigotes. However, Tc24 is then upregulated in association with parasite flagellar growth linked to re-transformation into the trypomastigote form, prior to host cellular escape. These observations are discussed in the context of potential mechanisms of vaccine immunity.

## Author summary

Chagas disease is a chronic infection with *Trypanosoma cruzi* (*T. cruzi*) that affects approximately 8 million people worldwide and may cause chronic heart inflammation. The vaccine candidate Tc24-C4 is a recombinant version of the Tc24 protein, which is a flagellar calcium-binding protein expressed by *T. cruzi*. While animal challenge studies have shown that targeting Tc24 is very promising, it is not fully understood how Tc24 is presented to the immune system. Here, we were able to localize Tc24 in flagellated *T.

**Data Availability Statement:** Complete data can be found here: 10.6084/m9.figshare.14219723 10.6084/m9.figshare.14214335 10.6084/m9.figshare.14214323.

**Funding:** This work was funded by the Carlos Slim Foundation. (JP, LV, RA, CP, JVM, KMJ, PJH, MEB). This project was supported by the Protein and Monoclonal Antibody Production Shared Resource at Baylor College of Medicine with funding from NIH Cancer Center Support Grant P30 CA125123. The operation of the Cytometry and Cell Sorting Core at Baylor College of Medicine was funded by the CPRIT Core Facility Support Award (CPRIT-RP180672) and the NIH (CA125123 and RR024574). Molecular graphics and analyses were performed with UCSF Chimera, developed by the Resource for Biocomputing, Visualization, and Informatics at the University of California, San Francisco, with support from NIH P41-GM103311. The funders had no role in study design, data collection and analysis, decision to publish, or preparation of the manuscript.

**Competing interests:** I have read the journal's policy and the authors of this manuscript have the following competing interests: Leroy Versteeg, Rakesh Adhikari, Cristina Poveda, Maria Jose Villar-Mondragon, Kathryn M Jones, Peter J Hotez, Maria Elena Bottazzi and Jeroen Pollet are involved in the development of a vaccine against Chagas disease.

*cruzi* parasites using a novel Tc24-specific monoclonal antibody. The results showed that Tc24 is not exposed on the outside of the parasite, which suggests that antibodies against Tc24 could not bind parasites during the infection. Then, by analyzing Tc24 expression in T. *cruzi*—infected host cells over time, we observed that Tc24 expression is reduced after the parasite enters the cells but is restored when parasites escape the host cell again. Our study provides more insights on the location and presence of Tc24 in *T. cruzi* during infection in the host, and we discuss our current understanding on the mechanisms of how the Tc24 vaccine may work.

## Introduction

Chagas disease is a neglected tropical disease caused by the protozoan *Trypanosoma cruzi*. Approximately 6–8 million people are infected, with the highest prevalence in Latin America. [1] From individuals who become chronically infected with the disease, 30–40% develop cardiomyopathy, arrhythmias, and megaviscera.[2] There are only two anti-trypanosomal drugs, nifurtimox and benznidazole, which are licensed to treat Chagas disease. Because both drugs have severe adverse side effects and exhibit low efficacy in the chronic phase of infection, there is an urgent need for alternative, complementary or more effective treatments.[3,4] Prophylactic and therapeutic vaccines are considered potential immune strategies to control *T. cruzi* infection and/or progression of disease.[5]

A promising vaccine candidate antigen is the 24-kDa flagellar calcium-binding protein (FCaBP) of *Trypanosoma cruzi*. FCaBP is an immunogenic protein that is located in the flagellum of *T. cruzi*. Low levels of polymorphism of the gene encoding FCaBP suggest that this can be an effective vaccine candidate against multiple *T. cruzi* strains.[6] FCaBP comprises four EF-hand calcium-binding motifs, of which the third and fourth are able to bind calcium.[7] While the exact function is yet to be elucidated, it is hypothesized that FCaBP acts as a calcium sensor and is involved in regulating $Ca^{2+}$ dependent cell signaling pathways in the parasite.[8] In the Chagas vaccine field FCaBP, in this field commonly known as Tc24,[9] was shown to have immunoprotective properties in a BALB/c acute lethal mouse model.[10] The antigen was further explored as a DNA vaccine in dogs,[11] and as a recombinant protein nanoparticle vaccine in mice.[12] A suitable platform was developed for the large scale production of recombinant Tc24 [13] and Tc24 was selected as one of the key antigens under consideration for a human therapeutic Chagas disease vaccine[14] supported by multiple preclinical studies with a recombinant Tc24 vaccine. [12,15,16]

As previously published, to prevent aggregation of recombinant Tc24 during the production process, four cysteine codons were replaced by serine codons. The resulting antigen, designated Tc24-C4, showed less aggregation while secondary structure and immunogenicity was not altered, and the production process was found to be suitable for technology transfer in preparation for its production under current Good Manufacturing Practices (cGMP).[17,18] It was further shown in a mouse model that vaccination with Tc24-C4 improved the efficacy of benznidazole treatment and reduced myocarditis and fibrosis during acute *T. cruzi* infection.[19,20]

*T. cruzi* has a complex life cycle that involves two different stages of the parasite during infection in the vertebrate host.[21] Trypomastigotes are the parasitic stage with developed flagella that can be found in the bloodstream and in the extracellular spaces of the host. Once trypomastigotes enter a host cell, they discard their long flagella, and transform to the amastigote stage and a truncated flagellum remains.[22,23] They then divide several times by binary

fission. Following division, the amastigotes transform back to trypomastigotes, exhibiting continuous flagellar movement. Eventually, the host cell wall ruptures and trypomastigotes are released in the extracellular space and bloodstream.[24]

Revealing the location and the presence of Tc24 in the different stages of the *T. cruzi* parasites may help explain the protection mechanism of Tc24 as a vaccine antigen. It was previously hypothesized that Tc24 is located in the flagellar pocket of the parasite,[13,25,26] which would suggest that antibodies could bind to the trypomastigotes, possibly preventing cell invasion. However, in the broader field of trypanosomatids research, it has been shown that flagellar calcium-binding proteins are typically localized intracellularly on the flagellar membrane, [27,28] specifically anchoring to the inner leaflet of the flagellar membrane.[7,29–31] In this case, a humoral response would unlikely be effective to prevent a trypomastigote from infecting host cells. Here, we developed and described a new monoclonal antibody (mAb) with specificity to Tc24-C4, which was used to localize native Tc24 in *T. cruzi* by different microscopic techniques. Fluorescence confocal microscopy and imaging flow cytometry revealed that Tc24 is expressed intracellularly, and is stage-specific amplified in association with a transformation from the amastigote to trypomastigote form and released by cellular rupture. The implications for *T. cruzi* vaccine mechanism are further discussed.

## Materials and methods

### Ethics statement

Animal studies were approved by the Institutional Animal Care and Use Committee of Baylor College of Medicine and were performed in strict compliance with The Guide for Care and Use of Laboratory Animals (8th Edition).[32]

### *T. cruzi* strain

*T. cruzi* MHOM/MX/0000/H1 (H1) strain was used for all experiments. This strain was originally identified in the Yucatán area of Mexico, and is classified as Discrete Typing Unit (DTU) TcI.[33,34]

### Production of Tc24-WT and Tc24-C4

Tc24-WT and Tc24-C4 were produced according to previously published methods.[17] Briefly, DNA coding for Tc24-WT and Tc24-C4 was codon-optimized, chemically synthesized and cloned into the *E. coli* expression vector pET41a and transformed into *E. coli* BL21 (DE3). For the Tc24-C4 construct, all four cysteines in the Tc24-WT sequence were replaced by serine residues. For recombinant protein expression, 10 L of BSM media in a fermentor was inoculated with the transformed *E. coli*. After the media reached the desired cell density, IPTG was added to induce the Tc24-WT expression. After the fermentation, the biomass was collected and homogenized using an EmulsiFlex C3 (Avestin, Canada) (for Tc24-WT) or EmulsiFlex C-55 (Avestin, Canada) (for Tc24-C4). The extracted proteins were further purified using Q Sepharose XL columns and size-exclusion chromatography (SEC) methods.

### Preparation of *T. cruzi* lysate

Lysate from *T. cruzi* trypomastigotes and amastigotes was made using a previously published method.[35] The lysate was prepared without any detergents to keep the protein structures stable. To prepare trypomastigotes, VERO cells were infected with *T. cruzi* trypomastigotes at a multiplicity of infection (MOI) of 5 and incubated at 37°C with 5% $CO_2$. After 5 days, the trypomastigotes were harvested from the culture supernatant. To prepare amastigotes, infected

cells were infected for 6 hrs before extracellular (trypomastigote) parasites were removed. Cells were further incubated for an additional 48 hrs, followed by removal from the flask using Accutase (Innovative Cell Technologies, Cat# AT104). Cells were centrifuged for 5 min at 300 x g and resuspended in PBS. To release the intracellular amastigotes, the cells were transferred to a gentleMACS M Tube (Miltenyi Biotech, Cat# 130-093-236) and dissociated using a gentle-MACS Octo Dissociator (Miltenyi Biotech) following protocol "Protein_01_01".[36] Afterwards 1.5 mL fractions of the dissociated cell material were loaded onto pre-equilibrated PD-10 columns to separate the intracellular amastigotes from cell debris according to previous published methods.[37] Additional PBS was added to the top of the column, while 1 mL fractions of eluted material were collected. Following visual assessment under a light microscope, the intracellular amastigote–containing fractions were pooled and pelleted by centrifugation at 3,000 x g for 5 min. Two washes of the intracellular amastigotes were performed using centrifugation and resuspension in PBS to remove VERO cell debris.

To create the lysate, purified parasites were disrupted by three freeze/thaw cycles in PBS containing protease inhibitor cocktail (cOmplete ultra-tablets, Roche). Parasite lysate was then sonicated three times for 15 s each and subsequently centrifuged for 30 min at 15,000 x g. The protein concentration of the soluble fraction was determined by using a BCA protein assay kit (ThermoFisher, Cat# 23225). The final *T. cruzi* lysate samples were stored at -80˚C until use.

## Development of Tc24-C4 specific B-cell hybridoma's

Female BALB/c mice of 6–8 weeks old were immunized three times intraperitoneally (i.p.), two weeks between immunizations, with 100 μg Tc24-C4 + Freund's complete adjuvant as first immunization and 50 μg Tc24-C4 + Freund's incomplete adjuvant for booster immunizations. One week after the third immunization, the mice were bled via retro-orbital sinus puncture and titers were determined by indirect ELISA and western blot. The mouse with the highest serum reactivity in ELISA and western blot was selected to perform the fusion. The mouse received a final i.p. boost with 100 μg of Tc24-C4 and five days after the final boost, mice were humanely euthanized and spleens were harvested. Splenocytes were obtained by grinding the spleen through a steel mesh screen to generate a single-cell suspension. A fusion between splenocytes from the chosen Tc24-C4 immunized mouse and the mouse SP2/0 myeloma cell line was performed using standard PEG fusion methodology. Newly formed hybridomas from the fusion were plated in ClonaCell Medium D (StemCell Technologies, Inc.), a semi-solid methylcellulose-based selection media, and allowed to grow for twelve days prior to identifying, picking, and transferring individual hybridoma clones to wells of 96-well tissue culture plates using the ClonaCell Easy Pick robot (StemCell Technologies). The hybridomas were grown for three days and supernatants of hybridomas were screened by indirect ELISA for reactivity to Tc24-C4. Those hybridomas showing a strong ELISA reaction were transferred to 24-well tissue culture plates and the supernatants were screened again by ELISA and Western blot for reactivity to Tc24-C4. Hybridoma clone #884 was selected and adapted to IMDM culture medium + 15% fetal bovine serum (FBS) and cryopreserved in liquid nitrogen.

## Production and purification of Tc24-C4 specific monoclonal antibodies

The B cell hybridoma clone #884 was thawed and seeded at 100,000 cells/mL in 10 cm diameter suspension culture dishes in a volume of 20 mL. Hybridomas were grown for 8–10 days in 15% FBS with ultra-low IgG (ThermoFisher, Cat# 16250078) in IMDM culture media with addition of 2–3 mL new culture media every 3–5 days. Once the viability of the clones became lower than 50%, the culture supernatant containing the secreted antibodies was harvested. The cells were spun down at 300 x g for 5 min, and the supernatant was stored at 4˚C. A Pellicon

XL50 with Ultracel 30 kDa Membrane, C screen, 50 cm$^2$, and Labscale TFF system was used for concentration of the supernatant and buffer exchange to 20 mM of $NaH_2PO_4$ + 20mM of $Na_2HPO_4$ solution. The antibodies were then individually purified using HiTrap Protein G HP Columns (GE, Cat#. GE17-0404-01). Approximately 8 mg of mAb Tc24-C4/884 was obtained from 400 mL of culture supernatant.

### Enzyme-Linked ImmunoSorbent Assays (ELISAs)

Plates were coated with 100 µL 0.3125 µg/mL recombinant Tc24-C4 or Tc24-WT (produced in *E. coli*) in 1X KPL coating solution (Sera Care, Cat# 5150–0014) and incubated overnight at 4˚C. For coating of the *T. cruzi* lysate a concentration of 10 µg/mL was used. The next day plates were tapped dry and blocked overnight with 200 µL/well 0.1% BSA in 1x PBS + 0.05% Tween-20 (PBST). The next day coating solution was decanted, plates were sealed and stored at -20˚C until further use. mAb Tc24-C4/884 was added at a starting concentration of 20 µg/mL and diluted two-fold across. After two hrs, plates were washed 4x with PBST using a plate washer. Goat anti-mouse IgG (H+L) antibodies conjugated with HRP (LSBio, Cat# LS-C55886) were diluted 4000 times in 0.1% BSA in PBST, and subsequently 100 µL was added per well. Plates were incubated for 1 hour. After 1 hour of incubation, the plates were washed 5x with PBST using the plate washer. Next, 100 µL SureBlue TMB substrate (SeraCare, Cat# 5120–0077) was added per well. Plates were developed in the dark at room temperature, and the reaction was stopped after exactly 4 min. using 100 µL 1 M HCl. The optical density at 450 nm ($OD_{450}$) was measured using a spectrophotometer (Epoch 2, BioTek).

To determine the isotype of mAb Tc24-C4/884, plates were coated overnight at 4˚C with 0.1 µg/mL with the mAbs. The next day plates were tapped dry and blocked overnight with 200 uL/well 0.1% BSA in PBST at 4˚C. The following day goat anti-mouse IgG1, IgG2a and IgG2b antibodies conjugated to HRP (LSBio Cat # LS-C346714-1, Cat# LS-C346721-1 and Cat# LS-C346730-1 respectively) were diluted 4,000 times in 0.1% BSA in PBST and 100 µL each antibody was added to another well. After 1-hour incubation, the plates were washed 5x with PBST using the plate washer. SureBlue TMB substrate was removed from 4˚C refrigerator and 100 µL was added per well. Plates were developed in the dark, and the reaction was stopped at exactly 4 min. using 100 µL 1 M HCl. The absorbance at 450 nm was measured using a spectrophotometer (Epoch 2, Biotek).

### Western blots and Coomassie Brilliant Blue staining

Reduced and non-reduced Tc24-C4 (0.25 µg), Tc24-WT (0.25 µg) and *T. cruzi* trypomastigote lysate (2 µg) were loaded on 4–12% Bis-Tris gels together with a SeeBlue Plus2 marker (Thermo Fisher Scientific). After running for approximately 2 hrs at 100 V, the gel was transferred to western blots using an iBlot 2 Dry Blotting System (Thermo Fisher Scientific). Western blot was soaked in methanol for one minute and subsequently blocked for 2 hrs with 5% non-fat dry milk in PBST. The western blot was then incubated overnight with 1 µg/mL of mAb Tc24-C4/884 in 1% non-fat dry milk in PBST. After incubation, the Western blot was washed 4 times with PBST followed by 1-hour incubation with 1:5000 diluted goat anti-mouse IgG alkaline phosphatase (KPL, Cat# 0751–1806) in PBST. After incubation, it was washed 4 times with PBST and 1 time with PBS. Finally, immunodetection was performed using BCIP/NBT substrate (VWR, Cat# 50-81-00).

To compare the expression of Tc24 in trypomastigotes to the expression in amastigotes, we loaded Tc24-C4 (0.5 µg), *T. cruzi* trypomastigote lysate (1.5 µg) and *T. cruzi* amastigote lysate (1.5 µg) on two 4–12% Bis-Tris gels as described above. One gel was used for western blotting to detect Tc24 in the lysate samples, the other gel was stained with Coomassie Brilliant Blue to

show that equal an amount of trypomastigote and amastigote lysate material was loaded on the gels.

### Linear epitope prediction assays

Linear epitope prediction for mAb Tc24-C4/884 was performed using PEPperMAP Linear Epitope Mapping (PEPperPRINT).[38] Briefly, peptides of 15 amino acids length were synthesized, covering the theoretical amino acid sequence of Tc24-C4, including GSGSGSG linkers added to the C- and N-terminus. Each peptide had an overlap of 14 amino acids between neighboring peptides. This resulted into 211 unique peptide sequences, which were printed on a microarray plate in duplicate. Control peptide sequences HA (YPYDVPDYAG) and c-Myc (EQKLISEEDL) were also included. The microarray was incubated with 1 or 10 µg/mL Tc24-C4/884 followed by staining with goat anti-mouse IgG (H+L) DyLight 680 and mouse mAb anti-HA (12CA5) DyLight 800. Plate readout was performed using an LI-COR Odyssey Imaging System at 680 nm (red) and 800 nm (green).

### Competitive ELISA

The specificity of Tc24-C4/884 to the epitope TAEAKQR(R) was confirmed using a competitive ELISA. Four different peptides were purchased (GenScript Biotech) which included 1) TAEAKQRR: the expected epitope, 2) PREKTAEAKQRRIEL: the expected epitope with the flanking peptide sequences used in the linear epitope prediction assay (PEPperPRINT), 3) RIRQAIPREKTAEAK: peptide sequence containing the partial expected epitope sequence TAEAK, and 4) PAALFKELDKNGTGS: a randomly selected peptide sequence from Tc24 closer to the C-terminus of the protein. Peptides were serially diluted and pre-incubated with Tc24-C4/884, allowing the peptides to bind to the mAb. The tested molar ratios between peptide: Tc24-C4/884 started at 125:1 and continued in a two-fold dilution fashion until ratio 0.002:1 was reached. After one hour, mAb–peptide samples were transferred to a Tc24-C4 – coated ELISA plate and samples were incubated for two hrs. Following 4 washes with PBST, bound mAb was detected by goat anti-mouse IgG (H+L) antibodies conjugated with HRP (LSBio, Cat# LS-C55886). After one hour incubation, followed by 5 washes with PBST, Sure-Blue TMB substrate (SeraCare, Cat# 5120–0077) was used for signal development. After 1 M HCl was used to stop the color reaction, the optical density at 450 nm (O.D.450) was measured using a spectrophotometer (Epoch 2, BioTek).

### Tc24-C4 protein primary sequence alignment

To check for the conservation of the epitope of mAb Tc24-C4/884 between different *Trypanosoma* species, primary protein sequences of Tc24 were compared to each other. The protein sequence of the flagellar calcium-binding protein (protein entry P07749, UniProt.org) was used to perform a BLAST search to find similar protein sequences. The BLAST result file was screened for the epitope TAEAKQR(R). The protein sequences that contained the epitope were aligned using the Align tool on UniProt.com. FCaBP primary sequences of *T. brucei* and *T. Congolense* were added as a reference in S4 Fig.

### Fluorescence confocal microscopy

*T. cruzi* trypomastigotes were thawed and resuspended in fixation buffer (BD Biosciences) to fixate parasites, or in fixation/permeabilization buffer (BD Biosciences, Cat# 554714) to fixate and permeabilize parasites. Poly-L-Lysine—coated glass slides were placed in 6-well plates and resuspended trypomastigotes were added to the wells. The plate was then centrifuged for 20

min at 1500 x g at 4˚C to coat the trypomastigotes to the glass slides. Then the fix or fix/perm buffer was removed and trypomastigotes were washed 3 times with 2% FBS in PBS (staining buffer) for fixed parasites or 1 x perm/wash buffer for fixed/permeabilized parasites. Trypomastigotes were then incubated with either 10 μg/mL mAb Tc24-C4/884, 1:1000 diluted Tc24-C4 polyclonal mouse antisera, or 1:1000 diluted naïve mouse antisera. After 1 hour at 4˚C trypomastigotes were washed 3 times followed by an incubation 4˚C with 10 μg/mL anti-mouse IgG Alexa Fluor 488 antibody. After 1 hour, parasites were washed 4 times with assay buffer and subsequently incubated for 10 min with 1 μM DAPI to stain the nuclei. Finally, after 2 washes with assay buffer glass slides were mounted on microscopy slides and dried overnight before microscopy analysis.

For additional fluorescence confocal microscopy analysis of *T. cruzi* infected cells, VERO cells (ATCC, Cat# CCL-81) and mouse primary cardiac fibroblasts (Cellbiologics, Cat# C57-6049) were used. VERO cells are commonly used for *in vitro T. cruzi* infection studies,[39] and mouse primary cardiac fibroblasts were used since they are a natural cell target for *T. cruzi in vivo*.[40] VERO cells and mouse primary cardiac fibroblasts were grown on a Poly-L-Lysine—coated glass slides and infected with *T. cruzi* trypomastigotes at a MOI of 3. After 4 days, cells were fixed and permeabilized with BD fix/perm solution (BD Biosciences) followed by staining steps. For the VERO cells staining of native Tc24 was performed using mAb Tc24-C4/884 followed by goat anti-mouse IgG (H+L) Alexa Fluor 488 as a secondary antibody. DAPI was used to stain the nuclei, and CyTRAK Orange (eBioscience) to stain the DNA and cytoplasm. For the infected mouse primary cardiac fibroblasts, staining of the parasites was performed using an amastigote-specific surface protein (SSP4)-specific mouse IgG1 antibody and a secondary anti-mouse IgG Alexa Fluor 488 antibody. Staining of native Tc24 was performed using a biotinylated mAb Tc24-C4/884 using streptavidin–Alexa Fluor 555 as a secondary antibody. Filamentous actin was stained used phalloidin–iFluor 647, and the nucleus of the cardiac fibroblasts and parasites were stained using DAPI. The sample was analyzed using a Leica SP8 system and an HC PL APO 63x/1.40 oil objective was used. Post-processing of the image was performed using Fiji (ImageJ) and LAS X (Leica).

## Imaging flow cytometry

VERO cells were grown in 6-well culture plates and incubated with *T. cruzi* trypomastigotes at a MOI of 3. After six hrs the trypomastigotes were washed off, and cells were further incubated at 37˚C. At 6 hrs, 12 hrs, 24 hrs, 48 hrs and 96 hrs post infection, cells were removed from the 6-well plate using Accutase (Innovative Cell Technologies, Cat# AT104) and stained for Tc24 and SSP4. Briefly, cells were fixed and permeabilized with BD fix/perm solution (BD Biosciences, Cat# 554714) followed by staining using an amastigote-specific surface protein (SSP4)-specific mouse IgG1 antibody (BEI Resources, Cat# NR-50892) and a secondary anti-mouse IgG Alexa Fluor 488 antibody. Staining of native Tc24 was performed using a biotinylated mAb Tc24-C4/884, using streptavidin–Alexa Fluor 647 as a secondary antibody. Nuclei of the VERO cells were stained using DAPI. Cells were then analyzed using an Amnis ImagestreamX Mark II (Luminex, Austin) and images were analyzed and processed using IDEAS software. To compare the expression of Tc24 and SSP4 between the different time points, threshold masks Threshold(M11, Ch11 Tc24, 60) and Threshold(M02, Ch02 SSP4, 80) were created and used to create features Mean Pixel Threshold (M11, Ch11 Tc24, 60) _Ch11 Tc24 and Mean Pixel Threshold (M02, Ch02 SSP4, 80)_Ch02 SSP4. The Mean Fluorescent Intensity (MFI) of the complete populations was calculated and plotted in a histogram.

## Results

### Characterization of Tc24-C4/884 monoclonal antibody

**mAb Tc24-C4/884 binds to recombinant Tc24-C4 and Tc24-WT, and native Tc24 in *T. cruzi* lysate.** To evaluate whether the monoclonal antibody Tc24-C4/884 generated against recombinant Tc24-C4 protein and produced in *E. coli* could recognize recombinant Tc24-C4, Tc24-WT as well as native Tc24, different concentrations of the monoclonal antibody were used in an ELISA assay with plates coated with either Tc24-C4, Tc24-WT or a *T. cruzi* lysate (as a source of native Tc24).

The results in Fig 1A show that mAb Tc24-C4/884 detects Tc24-C4, Tc24-WT and native Tc24 in *T. cruzi* lysate. The results show the highest $OD_{450}$ signal for Tc24-WT and Tc24-C4, and a lower $OD_{450}$ signal for *T. cruzi* lysate. The signal for Tc24-WT is higher than for Tc24-C4, which might be caused by disulfide bond aggregation of Tc24-WT, leading to more coated molecules in the plate. The relatively low $OD_{450}$ signal in ELISA on the *T. cruzi* lysate compared to the signals obtained with a coat of Tc24-C4 or Tc24-WT, can be explained by the low concentration of native Tc24 in *T. cruzi* lysate and the excess of other *T. cruzi* proteins in the lysate that compete with Tc24 for binding to the ELISA plate. Therefore, the amount of native Tc24 that actually bound to the ELISA plates is likely to be much lower than for the purified recombinant proteins Tc24-C4 and Tc24-WT.

Furthermore, antibody isotyping of mAb Tc24-C4/884 revealed that this antibody is part of the IgG1 class (see S1 Fig).

After examining the binding of mAb Tc24-C4/884 against different versions of Tc24 on ELISA, it was further evaluated whether the mAb could detect different forms of Tc24 on a western blot. In Fig 1B it was shown that Tc24-C4/884 detects SDS-denatured reduced and non-reduced Tc24-C4, Tc24-WT and native Tc24 from *T. cruzi* lysate. Since there was no significant difference observed in band intensity between denatured reduced and non-reduced versions of Tc24, this suggests that Tc24-C4/884 is specific against a linear epitope.

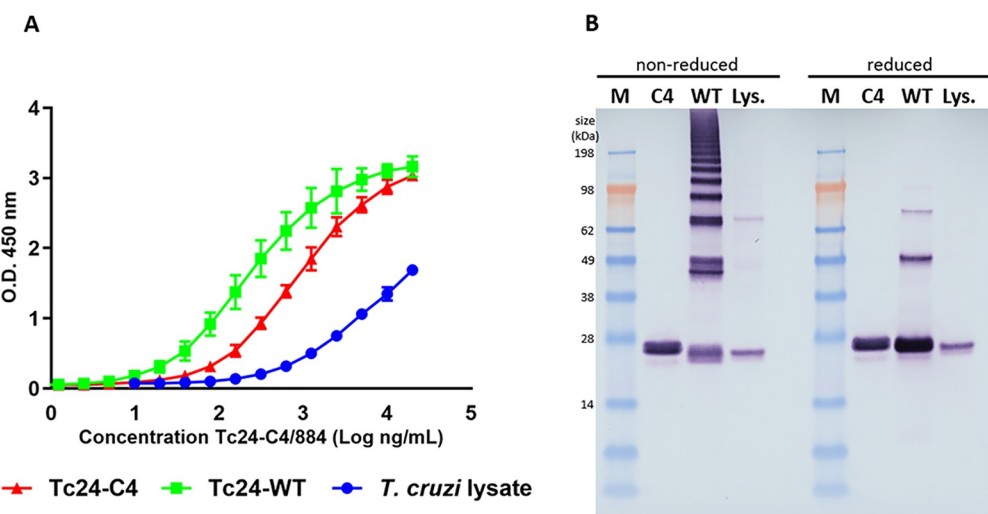

**Fig 1. mAb Tc24-C4/884 recognize Tc24-C4, Tc24-WT and native Tc24 in *T. cruzi* lysate in ELISA and Western blot.** A) The binding of mAb Tc24-C4/884 to Tc24-C4 (red), Tc24-WT (green) and native Tc24 in *T. cruzi* trypomastigote lysate (blue) by ELISA. All ELISAs were performed in triplicate. B) Detection of reduced and non-reduced Tc24-C4, Tc24-WT and native Tc24 by Tc24-C4/884 by western blot. M: SeeBlue Plus3 marker. C4: Tc24-C4. WT: Tc24-WT, Lys.: *T. cruzi* trypomastigote lysate.

The difference in S-S bridge formation between Tc24-WT and Tc24-C4 can be observed in these western blots. Tc24-WT has four free cysteines, which allow the protein to form multimers under non-reducing conditions, as seen for Tc24-WT (Fig 1B). As previously published, when the four free cysteines are removed in Tc24-C4, the multimerization of the protein is strongly reduced.[17]

## Monoclonal antibody Tc24-C4/884 binds to linear epitope sequence TAEAKQR(R)

To investigate to which linear epitope mAb Tc24-C4/884 binds, linear epitope mapping was performed. Peptides of 15 amino acids lengths, with 14 amino acid sequence overlap between successive peptides, were printed on a microarray and binding of mAb Tc24-C4/884 at two different concentrations was tested. A positive signal was observed in a set of adjacent peptides at both concentrations, with stronger signals at the higher concentration of 10 µg/mL (Fig 2). Fig 2B shows the measured fluorescence intensity started to increase strongly at peptide sequence TAEKQRRIELFKKF and reduced again at sequence RQAIPREKTAEAKQR. These results show that the consensus sequence of the linear epitope recognized by Tc24-C4/884 is TAEAKQR(R). The last arginine (R) is not elemental, but the signal is significantly weaker without the last arginine added. The variation in binding of Tc24-C4/884 with the different peptides that all contain the consensus epitope can be due to structural (e.g. linear or helical) differences between the whole 15-mer peptides. Additionally, a competitive ELISA confirmed that the specificity of Tc24-C4/884 is the epitope TAEAKQR(R), since pre-incubation of Tc24-C4/884 with peptides containing the TAEAKQR(R) sequence reduced the binding of the mAb to Tc24 –coated ELISA plates, while peptides without the epitope sequence or with just that partial sequence did not (S2 Fig). The consensus sequence recognized by mAb Tc24-C4/884 maps in an alpha helix on the first EF-hand of Tc24 (S3 Fig).

To predict whether the epitope TAEAKQR(R) is conserved within different *Trypanosoma* species and if Tc24-C4/884 will probably detect those proteins, amino acid sequences from different *Trypanosoma spp.* were downloaded from the UniProt protein sequence database and aligned. This alignment shows that epitope TAEAKQR(R) is conserved within *T. cruzi* Cl Brener, *T. cruzi* Dm28c, *T. cruzi* marinkellei and *T. rangeli* (S4 Fig).

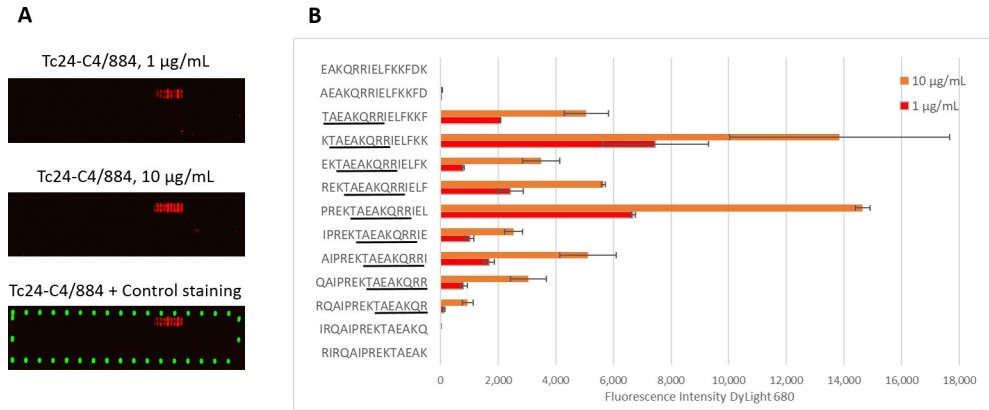

**Fig 2. Linear epitope mapping of monoclonal antibody Tc24-C4/884.** A) Scan of the microarray showing an increased intensity of DyLight 680 (red) in a stretch of adjacent peptide sequences when 1 and 10 µg/mL Tc24-C4/884 was used. DyLight 800 was used as a positive control (green). B) Measured median Fluorescent Intensity of DyLight 680 plotted with their corresponding peptide sequences. The consensus sequence is underlined in black.

## Studying the location and presence of Tc24 in *T. cruzi* parasites using the characterized antibody

**Tc24 is not exposed on the surface of *T. cruzi* trypomastigotes.**   To understand more about the location of Tc24 in *T. cruzi*, including the exposure of Tc24 on the surface, it was examined whether Tc24-C4/884 and Tc24-C4 antisera from Tc24-C4 –vaccinated mice could detect Tc24 in fixed only or in fixed and permeabilized *T. cruzi* trypomastigotes. Fig 3 shows that mAb Tc24-C4/884 binds strongly to *T. cruzi* trypomastigotes, which have been fixed and permeabilized, but not to trypomastigotes that were fixed but not permeabilized (fixed-only). Also, when trypomastigotes were incubated with pooled polyclonal anti-Tc24-C4 mice sera, IgG antibodies were only able to bind when trypomastigotes were fixed and permeabilized as well, confirming the observations seen with the mAb Tc24-C4/884. Pooled sera from naïve mice did not show IgG binding to fixed-only or fixed and permeabilized trypomastigotes. These results show that Tc24 is not exposed on the surface of the trypomastigotes but is present intracellularly.

**Tc24 is not equally expressed in different stages of *T. cruzi*.**   After it was found that mAb Tc24-C4/884 can detect native Tc24 in fixed and permeabilized trypomastigotes, it was tested if Tc24 could be detected during all stages of the infection of host cells. Green Monkey Kidney cells (VERO) were infected with *T. cruzi* trypomastigotes. After 72 hrs cells were fixed and permeabilized, and Tc24 was detected using Tc24-C4/884. Fig 4B shows the staining of mAb Tc24-C4/884 to Tc24 in the *T. cruzi* parasites at different stages of the parasite's life cycle, indicated by "I", "II", "III" and "IV". The morphological features of "I" indicates that this is either an extracellular or a recently internalized trypomastigote. The parasite stage labeled with "II"

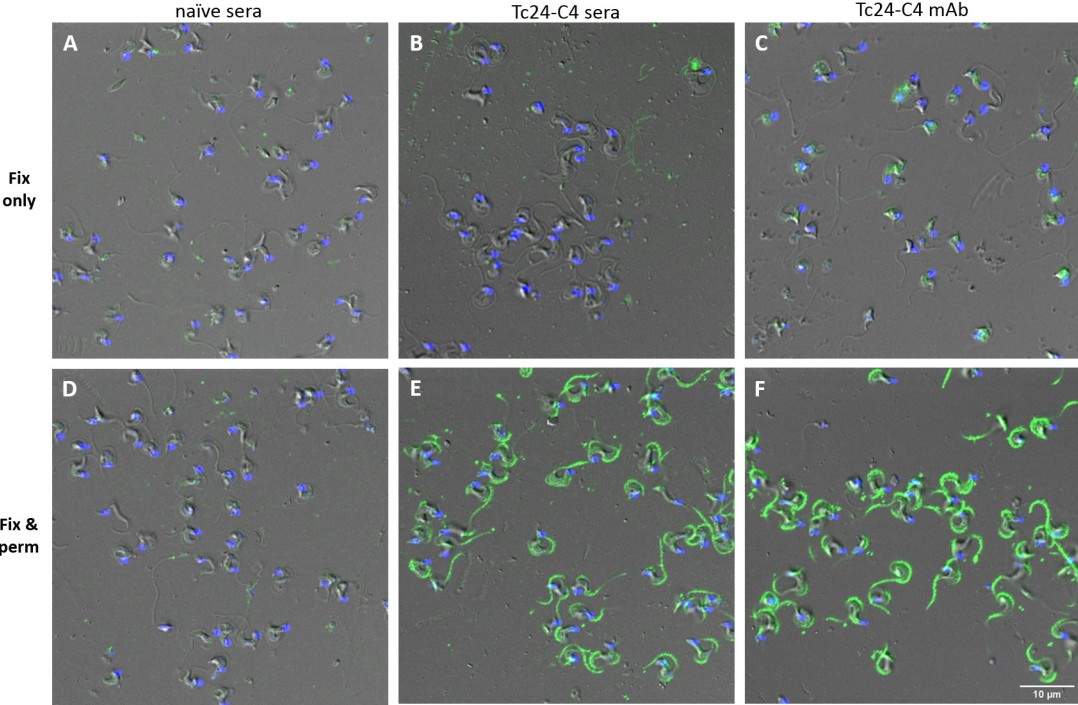

**Fig 3. Antibodies recognizing Tc24 binds strongly to fixed and permeabilized *T. cruzi* trypomastigotes but not to fixed-only trypomastigotes.** Staining of fixed-only (fix only) or fixed and permeabilized (fix & perm) *T. cruzi* trypomastigotes with mAb Tc24-C4/884, Tc24-C4 polyclonal antisera or naïve sera followed by staining with goat anti-mouse IgG (H+L) Alexa Fluor 488 (green). After staining DAPI (blue) was used to stain the nucleus and kinetoplast. Pixel size of images: 60 nm x 60 nm.

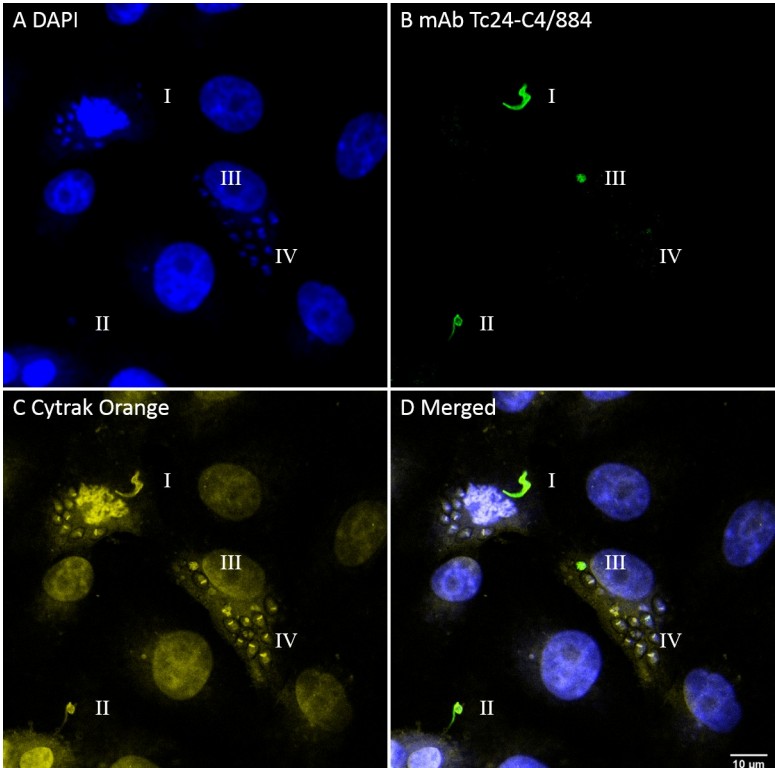

**Fig 4. Fluorescence confocal microscopy images of VERO cells infected with *T. cruzi* showing the different Tc24 expression in parasitic stages.** Different transitions between trypomastigote and amastigote can be observed in I, II, III and IV. I: trypomastigote, II: trypomastigote with reduced flagella, III: complete transition to amastigote, IV: amastigotes with multiple divisions. A) In blue, DAPI nuclear stain. B) In green, mAb Tc24-C4/884 stained by goat anti-mouse IgG (H+L) Alexa Fluor 488. C) In yellow, CyTRAK Orange DNA and cytoplasmic stain. D) Merged image of A-C. Pixel size of images: 102 nm x 102 nm.

has a round shape with reduced flagella, indicating it has been internalized for a longer period of time and is transforming to the amastigote stage. The single parasite labeled with "III" has lost its flagella and has the morphology of an amastigote but still lacks the disk-shaped kinetoplast which is distinct for replicating amastigotes.[41] The nuclear stain (DAPI) in Fig 4A and the DNA and cytoplasmic stain (CyTRAK Orange) in Fig 4C clearly shows the presence of replicating amastigotes in the cytoplasm of the infected cell (labeled "IV"), and is confirmed by the presence of disk-shaped kinetoplasts. However, these amastigotes do not show staining with mAb Tc24-C4/884 (Fig 4B). This strongly suggests that Tc24 expression is strongly reduced in amastigotes after multiple divisions inside a host cell.

To confirm the observation that Tc24 is not equally expressed in trypomastigotes and amastigotes, Tc24 in *T.cruzi* lysate from trypomastigotes and amastigotes was analyzed using western blotting (Fig 5B). Detection of Tc24 by mAb Tc24-C4/884 revealed a much stronger Tc24 band in the trypomastigote than in the amastigote lysate, further proving that Tc24 expression is reduced in amastigotes compared to trypomastigotes. A second SDS-PAGE gel loaded with the same samples was stained with Coomassie Brilliant Blue to show that a similar amount of lysate was loaded on the gels (Fig 5A).

**Expression of Tc24 is reduced in amastigotes but restores during transformation to trypomastigote stage.**    To further explore the expression of Tc24 during infection of *T. cruzi* in host cells, mouse primary cardiac fibroblasts were infected with *T. cruzi* and after 72 hrs the presence of Tc24 and an amastigote surface protein (SSP4) were examined. SSP4 is specific to

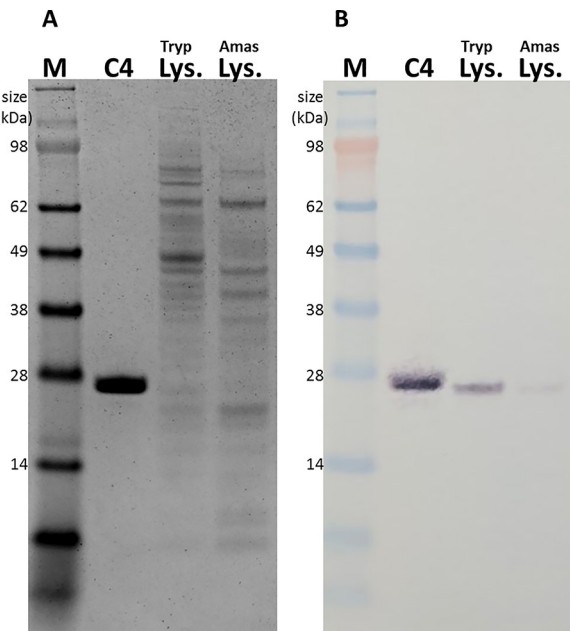

**Fig 5. Expression of Tc24 is reduced in *T. cruzi* amastigotes compared to *T. cruzi* trypomastigotes.** A) Coomassie Brilliant blue stained SDS-PAGE gel, and B) Detection of Tc24 in reduced Tc24-C4, *T. cruzi* trypomastigote lysate and *T. cruzi* amastigote lysate by Tc24-C4/884 using western blot M: SeeBlue Plus3 marker. C4: Tc24-C4. Tryp Lys.: *T. cruzi* trypomastigote lysate. Amas Lys.: *T. cruzi* amastigote lysate.

the amastigote stage, and is not expressed in trypomastigotes.[42] In Fig 6 two different stages of infected cardiac fibroblasts are shown. Fig 6A1–A5 are all from two infected cardiac fibroblasts and represent stage "IV" (like in Fig 4), and Fig 6B1–B5 show one infected cardiac fibroblast which represents stage "V". Stage "V" represents *T. cruzi* amastigotes that are transforming back to trypomastogites. Besides DAPI, Tc24 and SSP4, filamentous actin detection by Phalloidin was used to visualize the boundaries of each cell (Fig 6A4–B4). Merged images of all channels were given in Fig 6A5–B5.

Fig 6A1 shows *T. cruzi* amastigotes spread in the cytoplasm of the cell, while in the event of Fig 6B1 they are more concentrated around the nucleus. The cardiac fibroblast in Fig 6B1 is infected with more *T. cruzi* amastigotes than the cardiac fibroblasts in Fig 6A1, suggesting the fibroblast has been infected for a longer period of time. Interestingly, while the *T. cruzi* amastigotes stained very strongly for SSP4 (Fig 6A2), they stained dim and diffuse for Tc24 (Fig 6A3). On the contrary, in infected cardiac fibroblasts with many *T. cruzi* amastigotes, indicating continued replication of the parasite, much more expression of Tc24 was observed (Fig 6B3). Here *T. cruzi* amastigotes showed overall more staining for Tc24, and staining was even more increased around a structure that appears to be the growth of new flagella. This suggests that these amastigotes are transforming back to trypomastigotes.

The number of *T. cruzi* amastigotes in an infected cell increases quickly, since amastigotes divide by binary fission approximately every 18–25 hrs.[43] Therefore, it can be expected that the cardiac fibroblasts in Fig 6A was infected recently and only 2–4 amastigote divisions have happened. The stage of the amastigotes in Fig 6A is therefore very similar to the amastigotes at stage "IV" in Fig 4. In Fig 6B, the cardiac fibroblasts contained many amastigotes, so there the cell has been infected for a longer period of time and considerably more divisions took place. When an infected cell contains many amastigotes, indicative of continued divisions, the amastigotes start transforming back to trypomastigotes, which involves the growth of flagella

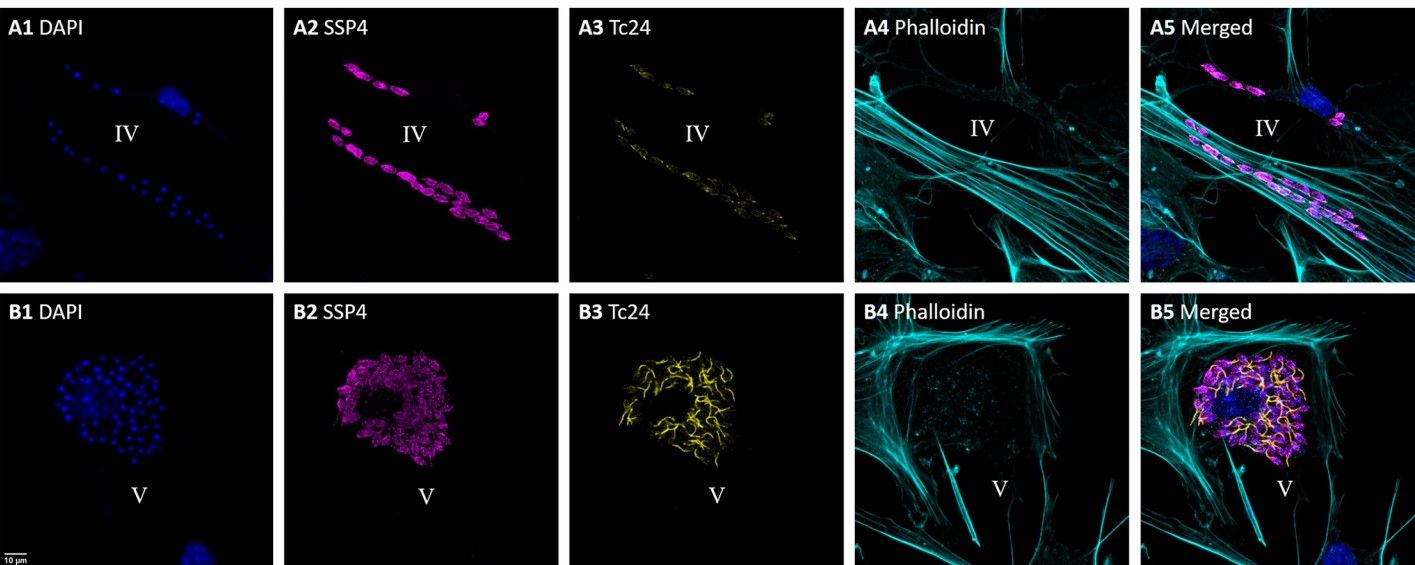

**Fig 6. Fluorescence confocal microscopy images of mouse primary cardio fibroblasts infected with *T. cruzi* show the return of Tc24 expression.** Two images were acquired that represent different stages of infection. In A1-A5 two infected cardiac fibroblasts can be seen with only approximately 20 and 6 amastigotes inside, representing stage IV: amastigotes with several (2–4) divisions. In B1-B5 an infected cardiac fibroblast can be seen hosting 60 amastigotes inside, which represents stage V: transformation from amastigote back to trypomastigote. 1) In blue, DAPI nuclear stain. 2) In magenta, detection of the amastigote specific protein SSP-4 using a specific IgG1 antibody followed by staining with anti-IgG1 Alexa Fluor 488. 3) In yellow, detection of Tc24 using biotinylated Tc24-C4/884 followed by straining with streptavidin–Alexa Fluor 555. 4) In cyan, filamentous actin stained with Phalloidin iFluor 647. 5) Merged image of 1–4. Pixel size of images: 120 nm x 120 nm.

(Fig 6B3 and 6B5). Fig 6B shows the moment when the transformation from amastigote of trypomastigote stage happens. This can be considered stage "V", which comes after stage "IV" shown in Fig 4.

**Expression of Tc24 is reduced after cellular invasion but increases prior to cellular escape.** To further confirm and quantify the expression of Tc24 and SSP4 by *T. cruzi* trypomastigotes and amastigotes, VERO cells were examined using imaging flow cytometry at different time points after infection. The cells were infected with *T. cruzi* trypomastigotes and subsequently fixed and stained for Tc24 and SSP4 at 3 hrs, 6 hrs, 12 hrs, 24 hrs, 48 hrs and 96 hrs after the start of the infection. In the first 24 hrs after infection, the expression of Tc24 is reduced while the expression of SSP4 is increased (Fig 7A). Expression of Tc24 is reduced at 12 hrs and 24 hrs compared to 3 hrs, the period in which the trypomastigote is transforming to an amastigote (Fig 7B). In the same period, SSP4 expression is increased when the trypomastigote transforms to an amastigote (Fig 7A and 7C). At 48 hrs, the amastigotes remain expressing similar levels of Tc24 and SSP4 as at 24 hrs. However, at 96 hrs there is a large increase in Tc24 expression, which is thought to be caused by the transformation of amastigotes back to trypomastigotes. Due to the combination of the high expression of Tc24 and the large amount of trypomastigotes in a host cell at this time point, individual parasites can no longer be distinguished and the whole host cell shows up red (Fig 7C, 96 hrs). Expression of SSP4 remained similar as at 48 hrs. Interestingly, by inspecting the individual images of the 96 hrs time point, a small subpopulation was identified which showed very high expression of Tc24 and no expression of SSP4. A manual gate was drawn (S5 Fig) to identify this subpopulation (Fig 7A and 7C, 96 hrs cellular escape). The fully reduced expression of SSP4 suggests that these parasites are completely transformed from amastigotes to trypomastigotes and will escape the host cell shortly. Since the host cell will rupture when the trypomastigotes escape, this population of cells will not accumulate over time. Overall, this dataset showed that Tc24 expression in

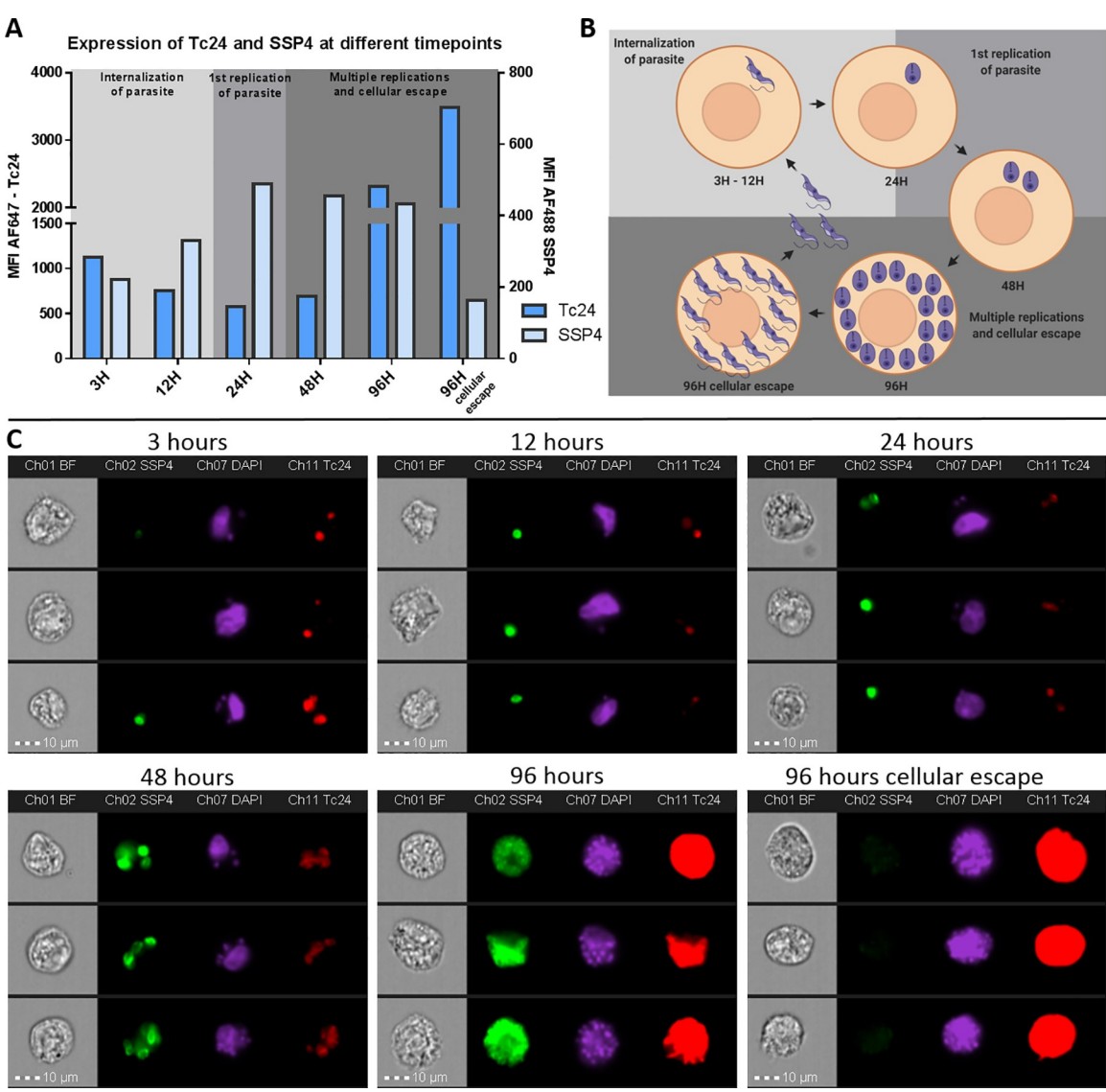

**Fig 7. Imaging Flow Cytometry reveals a change in expression of Tc24 and SSP4 during different infection timepoints of *T. cruzi*.**
VERO cells were infected with *T. cruzi* and 3 hrs, 6 hrs, 12 hrs, 24 hrs, 48 hrs and 96 hrs after start infection fixed and permeabilized. VERO cells were then stained for Tc24 and SSP4. Nuclei were stained using DAPI. A) The MFI of Tc24 and SSP4 from the total population of T. *cruzi*—infected VERO cells was plotted in a histogram. The left y-axis depicts the MFI for Tc24 and the right y-axis depicts the MFI for SSP4. B) A schematic representation of the transformation of *T. cruzi* trypomastigotes to amastigotes and back to trypomastigotes in the VERO host cell is shown. Prepared using Biorender.com C) Image gallery of three events of each infection time point representing the complete population. Presented are brightfield, Tc24 (red), SSP4 (green) and DAPI (magenta).

*T. cruzi* after cellular invasion decreases but expression increased again when the parasite transforms back to the trypomastigote stage.

## Discussion

We provide new information on the cell biology and localization of a flagellar calcium-binding protein called Tc24, and we discuss the underlying mechanisms linked to protective immunity for this antigen or its derivatives. The Tc24-C4/884 antibody recognizes a linear epitope in Tc24 and showed a strong affinity to recombinant Tc24-C4, Tc24-WT and native Tc24 in

*T. cruzi* lysate. This epitope is located on the first EF-hand of the Tc24 protein. Since only the third and the fourth EF-hand bind calcium,[7] it is anticipated that binding of Tc24-C4/884 does not interfere with calcium-binding by Tc24. Tc24 shows low diversity in protein sequence within *T. cruzi* stains[6]. By comparing the flagellar calcium-binding protein amino acid sequences of different *Trypanosoma spp*, it was found that the epitope recognized by Tc24-C4/884 is conserved between *T. cruzi* Cl Brener, *T. cruzi* Dm28c, *T. cruzi* marinkellei and *T. rangeli*. The conservation of the Tc24-C4/884 epitope in these *Trypanosoma* species suggests that the antibody can be used to detect Tc24 in these species, but the antibody cannot be used for a *T. cruzi*—specific serological test due to cross-reactivity with Tc24 from *T. rangeli*.

Images made by fluorescence confocal microscopy revealed that mAb Tc24-C4/884 as well as polyclonal Tc24-C4 antisera were only able to bind to Tc24 when *T. cruzi* trypomastigotes were fixed and permeabilized, but not when trypomastigotes were fixed only. This indicates that Tc24 is not exposed on the outside of trypomastigotes. This finding was somewhat surprising since it has previously been suggested that Tc24 is located on the outside of the *T. cruzi* parasites and antibodies against Tc24 can be generated that show complement-mediated trypanolytic activity.[10,44] The data supporting these suggestions in these manuscripts was however limiting; it was not tested whether antibodies against Tc24 could bind Tc24 without permeabilizing the parasite, and complement-mediated trypanolytic activity by Tc24 antisera was not shown to be significantly different from controls.[10]. Additionally, our microscopic analysis clearly shows that Tc24 is located along the whole flagellar membrane of trypomastigotes and is not concentrated in just the flagellar pocket, as earlier described.[13,18,26] Consequently, if Tc24 is not exposed on the surface of the parasite, antibody-mediated immune responses against Tc24 cannot opsonize the parasites and will not induce antibody-mediated phagocytosis nor anti-Tc24 mediated complement lysis of *T. cruzi*.

In-line with the current *T. cruzi* literature,[29] our images acquired by fluorescence confocal microscopy confirmed that Tc24 is expressed differentially during the trypomastigote and amastigote stages. Tc24 expression was high in the flagellum of trypomastigotes and reduced in replicating amastigotes, and this observation was confirmed by the much stronger detection of Tc24 in trypomastigote than amastigote lysate by western blotting. To add further, we used imaging flow cytometry analysis to quantify the decrease of Tc24 expression after *T. cruzi* infects a host cell and the increase at later time points in infection and cellular escape. While expression of Tc24 was found to decrease right after cellular invasion, SSP4 (surface marker of the amastigote stage) expression increased. Of note, in Fig 4B it looks like there is no expression of Tc24 in the replicating amastigotes (stage IV), while there is a dim Tc24 signal visible at 24H and 48H in Fig 7. This difference can be explained by the difference in imaging techniques. While confocal microscopy used for Fig 4 only collected emitted light from a thin slice of the sample, in imaging flow cytometry emitted light was collected from the complete depth of the sample. Therefore, more emitted light was acquired in the imaging flow cytometry experiment of Fig 7 and a dim signal of Tc24 could be observed at 24H and 48H post infection, while this was for these time points not visible by confocal microscopy (Fig 4).

*T. cruzi* flagellar-derived proteins like Tc24 are of special interest as vaccine candidates, since flagellar proteins are among the first proteins presented on the MHC of infected cells to CD8+ T cells.[22] Upon host cell entry, the *T. cruzi* flagellum reduces by approximately 90%, [45] gets "discarded" and flagellar-derived proteins become available to MHC class I processing and subsequent presentation to CD8+ T cells. Furthermore, the strain-invariant nature and abundance in flagellar-stage parasites of flagellar proteins make them very interesting vaccine targets. Immunization with paraflagellar rod proteins for example, have already shown to be protective against *T. cruzi* infection in mice through a T cell-dependent manner, while anti-paraflagellar rod antibodies do not bind to live parasites.[46–48] Similarly, a cytotoxic T cell

(CTL) response might be the key mechanism of Tc24 as a vaccine candidate, allowing for very effective elimination of *T.cruzi*–infected host cells and clear infection.[49,50]

The location and expression of Tc24 in *T. cruzi* trypomastigotes and amastigotes has direct consequences for Tc24 vaccine-induced protection. Due to the absence of Tc24 on the surface of the parasite, antibody-mediated immune response against Tc24 will not be able to opsonize the parasites and therefore also not to induce antibody mediated lysis or phagocytosis of the parasites. Our study does not entirely rule out the possibility of effector humoral immunity to Tc24. Because *T. cruzi* secretes Tc24 as soluble proteins and in extracellular vesicles (EVs), [51,52] it has been suggested that Tc24 might be involved in immune-evasion strategies or interference with host signaling pathways.[52,53] By inducing Tc24-specific antibodies in the host, the function of soluble Tc24 and Tc24 in EVs might be disabled and the protective immune response improved. Additionally, soluble Tc24 and Tc24 in EVs could be targeted by inducing catalytic antibodies that degrade Tc24 through IgM-mediated hydrolysis.[16,25] While it is yet to be elucidated what the effect of secreted Tc24 is on the host immune response, it is clear that Tc24 is an important target for cell-mediated immunity. Indeed, since *T. cruzi* is an intracellular pathogen most of its lifecycle, it is thought that a cytotoxic T cell response might be very effective in eliminating *T. cruzi*–infected host cell and clear the infection.[49,50] This is supported by the observation that antibody mediated depletion of CD8+ T cells in a *T. cruzi* infection model, resulted in increased susceptibility towards infection.[54] Villanueva-Lizama *et al.* showed that PBMC-derived effector memory T cells from chagasic patients produced IFN-γ and proliferated strongly after *in vitro* re-stimulation with Tc24.[55] Immunization with Tc24 vaccine candidates showed robust cell-mediated immune responses that had protective effects in mouse models[12,15,19] These studies suggest that a cell-mediated immune response is crucial to eliminate *T. cruzi* and clear the infection. Therefore, we hypothesized that the observed vaccine-induced protection by Tc24 immunization is mostly driven by cell-mediated immunity and Tc24 specific antibodies are unlikely to be directly involved in killing of *T. cruzi* parasites.

In summary, the results obtained using the monoclonal antibody Tc24-C4/884 yielded new insights into the location of the Tc24 protein and the expression of Tc24 in different stages of *Trypanosoma cruzi*. We found that Tc24 is not exposed on the surface of *T. cruzi* parasites. It is regulated in association with parasite-life cycle stages and transformation between amastigote and trypomastigote stages. Opportunities for Tc24-C4 vaccine-induced immunity include antibodies to soluble or EV bound Tc24 that might interfere with a possible immune-evasive function of Tc24, as well as a Th1-mediated response and CTLs clearing *T. cruzi*—infected cells.

## Supporting information

**S1 Fig. Indirect ELISA determined that the isotype of mAb Tc24-C4/884 was IgG1.** (TIF)

**S2 Fig. Competitive ELISA confirms Tc24/C4-884 is specific to epitope TAEAKQR(R).** Four peptides were pre-incubated with Tc24-C4/884 mAb at different molar ratios followed by binding of Tc24-C4/884 on Tc24-C4 –coated ELISA plates. A decrease in O.D. 450 signal for Tc24-C4/884 pre-incubated with peptides PREKTAEAKQRRIEL and TAEAKQRR suggest that Tc24-C4/884 mAb binding sites were blocked and that sequence TAEAKQR(R) is the specific epitope of Tc24-C4/884. (TIF)

**S3 Fig. Structural model showing that mAb Tc24-C4/884 binds to an epitope on the alpha-helix part of the first EF-hand of Tc24.** Epitope of Tc24-C4/884 is shown in red color. View

from the membrane-binding interface (left image) and a horizontal 180˚ rotation (right image). Protein structure of Tc24 was obtained from Protein Data Bank in Europe (PDB code 3CS1) and rendered using USCF Chimera. E: epitope of mAb Tc24-C4/884 (in red). EF1: residues 49–77 (in blue). EF2: residues 98–126 (in cyan). EF3: residues 131–159 (in blue). EF4: residues 168–196 (in cyan).30 C: C-terminus. N: N-terminus.
(TIF)

**S4 Fig. Primary sequence alignment of the *Trypanosoma cruzi* Tc24-C4 vaccine sequence with those of other Trypanosoma species shows that the epitope TAEAKQR(R) is conserved within *T. cruzi* Cl Brener, *T. cruzi* Dm28c, *T. cruzi* marinkellei and *T. rangeli*. UniProt sequence align tool (https://www.uniprot.org/align/).**
(TIF)

**S5 Fig. Gating strategy of imaging flow cytometry dataset to obtain the mean fluorescent intensity (MFI) of Tc24 and SSP4.** Events were first selected on focus using the gradient RMS feature followed by the selection of single cells only using the Area and Aspect Ratio features. The next gate involved the selection of cells which were infected by *T. cruzi*, which was achieved by gating around the cell population of an uninfected control sample. Finally, to remove background signal from "true" signal a Threshold mask was used for both Tc24 and SSP4 to select the area of interest to measure the MFI in. The features involving the Threshold mask for both Tc24 and SSP4 was plotted on a bivariate plot as shown in for 3 hrs, 48 hrs and 96 hrs. The MFI of the whole observed population was calculated and used in Fig 6A. By inspecting the images at 96 hrs, a manual gate was drawn which included events that expressed high MFI of Tc24 but no SSP4.
(TIF)

## Acknowledgments

We would kindly thank Dr. Kristyn Hoffmann for providing technical assistance with working with the mouse primary cardiac fibroblasts.

We would like to thank Dr. Alexandre Carisey from the Texas Children's Hospital William T. Shearer Center for Human Immunobiology for his expert assistance in the confocal microscopy experiments.

We would kindly thank Dr. Amal El-Mabhouh for providing technical assistance with analyzing the imaging flow cytometry dataset.

We especially like to mention Kurt Christensen and Karen Moberg for their support in the development of the mAbs.

## Author Contributions

**Conceptualization:** Leroy Versteeg, Kathryn M. Jones, Peter J. Hotez, Maria Elena Bottazzi, Edwin Tijhaar, Jeroen Pollet.

**Data curation:** Leroy Versteeg, Rakesh Adhikari, Cristina Poveda, Maria Jose Villar-Mondragon, Jeroen Pollet.

**Formal analysis:** Leroy Versteeg, Edwin Tijhaar, Jeroen Pollet.

**Funding acquisition:** Peter J. Hotez, Maria Elena Bottazzi.

**Investigation:** Leroy Versteeg, Maria Elena Bottazzi, Edwin Tijhaar, Jeroen Pollet.

**Methodology:** Leroy Versteeg, Kathryn M. Jones, Peter J. Hotez, Edwin Tijhaar, Jeroen Pollet.

**Supervision:** Kathryn M. Jones, Peter J. Hotez, Maria Elena Bottazzi, Edwin Tijhaar, Jeroen Pollet.

**Validation:** Leroy Versteeg, Edwin Tijhaar, Jeroen Pollet.

**Visualization:** Leroy Versteeg.

**Writing – original draft:** Leroy Versteeg.

**Writing – review & editing:** Leroy Versteeg, Rakesh Adhikari, Cristina Poveda, Maria Jose Villar-Mondragon, Kathryn M. Jones, Peter J. Hotez, Maria Elena Bottazzi, Edwin Tijhaar, Jeroen Pollet.

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
