## [Decision Letter · Decision Letter 0]

20 Apr 2021

Dear Dr. Pollet,

Thank you very much for submitting your manuscript "Location and expression kinetics of Tc24 in different life stages of Trypanosoma cruzi" for consideration at PLOS Neglected Tropical Diseases. As with all papers reviewed by the journal, your manuscript was reviewed by members of the editorial board and by several independent reviewers. In light of the reviews (below this email), we would like to invite the resubmission of a significantly-revised version that takes into account the reviewers' comments. 

We cannot make any decision about publication until we have seen the revised manuscript and your response to the reviewers' comments. Your revised manuscript is also likely to be sent to reviewers for further evaluation.

Sincerely,

Luisa Magalhães

Associate Editor

Ricardo Fujiwara

Deputy Editor

Reviewer's Responses to Questions

**Key Review Criteria Required for Acceptance?**

**Methods**

-Are the objectives of the study clearly articulated with a clear testable hypothesis stated?

-Is the study design appropriate to address the stated objectives?

-Is the population clearly described and appropriate for the hypothesis being tested?

-Is the sample size sufficient to ensure adequate power to address the hypothesis being tested?

-Were correct statistical analysis used to support conclusions?

-Are there concerns about ethical or regulatory requirements being met?

Reviewer #1: The methods are adequate and sufficient to complete the objectives of the study. Some suggestions from the reviewer are found in the attached review

Reviewer #2: The objectives of the study are clearly articulated with a clear testable hypothesis. The study design and the population are appropriate clearly described to address the stated objectives. The samples size is sufficient and the statistical analysis used support conclusions. There are no concerns about ethical or regulatory requirements being met.

Reviewer #3: The main goal of the work is to better characterize the expression and location of a T. cruzi vaccine candidate antigen, Tc-24, by using a monoclonal antibody. To my knowledge, all the experiments were carefully designed, with all the appropriate controls and statistical analysis, and mainly adequate to the purpose of the investigation: level of protein expression and localization of the protein within the vertebrate life cycle forms of the parasite, trypomastigotes and amastigotes. One experiment to validate level of protein expression could be performed to support the data. 

There are no concerns about ethical and regulatory requirements

**Results**

-Does the analysis presented match the analysis plan?

-Are the results clearly and completely presented?

-Are the figures (Tables, Images) of sufficient quality for clarity?

Reviewer #1: The results are well presented, some changes are suggested in the revision text.

Reviewer #2: The results are clearly presented and match the analysis plan.The figures have a good resolution and allow to identify the results. Although figure 3 allows to view the results, its resolution can be increased.

Reviewer #3: As mentioned, the experiments were adequately designed, as well as clearly presented (text and figures). There is a result from figure 4 there is not clear though. They show in the figure “diffrente” stages of amastigotes in the same cell, it was not clear how they determined this. Some other comments about the results and an experiment that should be performed to support data from protein expression levels between the two parasite stage life cycles are mentioned in the “summary and general comments”.

**Conclusions**

-Are the conclusions supported by the data presented?

-Are the limitations of analysis clearly described?

-Do the authors discuss how these data can be helpful to advance our understanding of the topic under study?

-Is public health relevance addressed?

Reviewer #1: The conclusions are adequate. Altough, two specific changes to the discussion are suggested

Reviewer #2: The conclusions are supported by the presented data. The limitations of the analysis can be better described and discussed in more detail. The authors discuss how the data can be helpful to advance our understanding of the topic under study.

Reviewer #3: The main conclusions are supported by the results. The authors also discuss how the data can be helpful to advance our knowledge. However, some points of the discussion should be reviewed or further addressed. This has been mentioned in the “summary and general comments”.

**Editorial and Data Presentation Modifications?**

Reviewer #1: (No Response)

Reviewer #2: The authors present many studies of Tc24 as a vaccine candidate in the introduction of the manuscript. In most of the discussion, the authors emphasize that the protein is inside T. cruzi, requiring permeabilization. This is really important, the main conclusion of the study. However, they do not discuss much about how it can affect immunization. The discussion regarding the immune response should be broader, addressing more work. The authors' results are good and may help in other studies, but they need to discuss further the Tc24 relationship and immune response.

Reviewer #3: Modifications or experimental suggestions are presented in the “summary and general comments”. No other minor modifications are necessary.

**Summary and General Comments**

Reviewer #1: This is a well conducted and presented study. It is suggested in the possible two experiments to demonstrate the specificity of the described epitope and a functional test with the monoclonal antibody in infection assays with cell cultures.

Reviewer #2: (No Response)

Reviewer #3: The main goal of the work was to better characterize the expression and location of a T. cruzi vaccine candidate antigen, Tc-24, by using a monoclonal antibody. All the experiments were carefully designed, with all the appropriate controls and statistical analysis, to my knowledge, to investigate the level of expression and localization of the protein within the vertebrate life cycle forms of the parasite, trypomastigotes and amastigotes. The main results of the manuscript are: the differential expression of Tc-24 between the two vertebrate life forms of the parasite and the fact that Tc-24 is not exposed to the outside of the flagellar membrane, which could question its use for a vaccine. However, some important points should be raised. 

- It has long been shown that Tc-24 is a flagellar calcium binding protein (Engman et al., J. Biol. Chem., 264, 18627–18631, 1989), linked to the membrane via calcium induced miristoylation and palmitoylation (Godsel and Engman, EMBO J. Apr 15;18(8):2057-65, 1999). Although Engman and coworkers, in the manuscript from 1989, had shown that Tc24 is expressed in all life forms of T. cruzi with different levels, here they show with different experimental approaches that its expression is reduced, at least in some stages of the amastigote form of the parasite. Nonetheless, this should be expected, since it had been previously shown, as mentioned, that Tc-24 is a flagellar calcium binding protein, mainly expressed in the flagellar membrane. As it is well known, amastigote forms present just a flagellar remnant. if the authors intended to prove that amastigotes really show reduced expression of Tc-24, it would be important to quantify the level of expression by performing a western blot of protein extracts of the two purified forms, using the Tc-24mAb. 

-In figure 4 the authors recon there are different amastigote stages on the same cell. However how they defined it or if this was only determined by Tc-24mAb labeling is not clear. The authors must make this point clearer in the text.

-The results from figures 4 and 6 are incongruent. On figure 4 they show that parasites that had just completed transitioned to amastigote still express Tc-24, its expression drastically decreases on fully differentiated dividing amastigote. On figure 6 the lower expression levels of Tc-24 is observed for intracellular parasites 24 hours post infection, which should correspond to trypomastigotes that had just transitioned to fully amastigotes. 48 and 96 hours later (when amastigotes are dividing), Tc-24 expression levels increases again. The same is true for the 96 hours post infection, even though at this time point there are also parasites that have transitioned back to trypomastigotes. The authors must revise this data or at least explain it better in the text and discus it further in the discussion section. Additionally, as mentioned before, if the authors intended to prove that amastigotes really show reduced expression of Tc-24, it would be important to quantify the level of expression by performing a western blot of protein extracts of the two purified forms, using the Tc-24mAb.

-In the manuscript from Godsel and Engman (1989) they had also shown that Tc24 is a member of the calcium–acyl switch protein family, therefore binding to the inner leaflet of the plasma membrane, which was corroborated later (Maric et al. J Biol Chem. 286(38): 33109–33117, 2011). Therefore, should be no surprise that labeling with Tc-24 mAB would only recognize Tc 24 upon fixation and permeabilization. Also in the discussion they mentioned that it has been previously suggested that the protein was located in the outside of the membrane, but this was based only in the fact that infection raises antibodies to this protein. However, as far as I am concerned, the first mention of this protein in the literature was as a member of the excretory/secretory product, so it is possible that this protein may be secreted when it is not bound to the flagellar membrane. In fact the authors even mention this in the discussion.

-Since the authors raise the point that this characterization is important for addressing the actual viability and role of this protein as a vaccine candidate, it would be important also to perform the labeling of purified proteins from parasite culture supernatant to verify whether it is really found as a secreted protein.

PLOS authors have the option to publish the peer review history of their article (what does this mean?). If published, this will include your full peer review and any attached files.

Reviewer #1: No

Reviewer #2: No

Reviewer #3: No
---

## [Decision Letter · Decision Letter 1]

27 Jul 2021

Dear Dr. Pollet,

We are pleased to inform you that your manuscript 'Location and expression kinetics of Tc24 in different life stages of Trypanosoma cruzi' has been provisionally accepted for publication in PLOS Neglected Tropical Diseases.

Best regards,

Luisa Magalhães

Associate Editor

Ricardo Fujiwara

Deputy Editor

Reviewer's Responses to Questions

**Key Review Criteria Required for Acceptance?**

**Methods**

-Are the objectives of the study clearly articulated with a clear testable hypothesis stated?

-Is the study design appropriate to address the stated objectives?

-Is the population clearly described and appropriate for the hypothesis being tested?

-Is the sample size sufficient to ensure adequate power to address the hypothesis being tested?

-Were correct statistical analysis used to support conclusions?

-Are there concerns about ethical or regulatory requirements being met?

Reviewer #1: The methodology of the manuscript is adequate and the data requested by the reviewer was completed. The objectives of the study are adequate for the methodology carried out.

Reviewer #3: As mentioned in the in the first review of the, all the experiments were carefully designed, with all the appropriate controls and statistical analysis, and mainly adequate to the purpose of the investigation.

Reviewer #4: Since the goal of the study is to explore the location and expression kinetics of Tc24 in different life stages of Trypanosoma cruzi, the methods are appropriate to verify the stated objectives. The analyzes are adequate to support conclusions and there are no concerns about ethical or regulatory requirements.

**Results**

-Does the analysis presented match the analysis plan?

-Are the results clearly and completely presented?

-Are the figures (Tables, Images) of sufficient quality for clarity?

Reviewer #1: The results obtained answer the questions of the researchers regarding the subject. An additional requested experiment was performed and complements the study. Pictures and tables are adequate.

Reviewer #3: All the points raised in the first review were addressed by the authors, including a new experiment performed to show the quantification of Tc24 in trypomastigote and amastigote forms, and are sufficient to sustain the conclusions presented in the manuscript. Also, some of the points raised were addressed in the discussion, which I believe improved the overall knowledge on the field

Reviewer #4: The results match the analysis plan, as well as they are clearly and completely presented, exploring relevant data related to the stated objectives. The figures have a sufficient quality for understanding the results.

**Conclusions**

-Are the conclusions supported by the data presented?

-Are the limitations of analysis clearly described?

-Do the authors discuss how these data can be helpful to advance our understanding of the topic under study?

-Is public health relevance addressed?

Reviewer #1: The conclusions of the study are supported by the results.

Reviewer #3: The main conclusions are supported by the results and, as mentioned, the discussion includes some important points that improved the overall knowledge on the field.

Reviewer #4: The conclusions are supported by the results presented and some limitations of analysis are described. The authors discuss how the data can be helpful to advance the understanding of the subject. The relevance to public health is addressed by presenting a possible vaccine candidate for the elimination of T. cruzi.

**Editorial and Data Presentation Modifications?**

Reviewer #1: This reviewer does not require additional modifications

Reviewer #3: no modifications are required at this time.

Reviewer #4: (No Response)

**Summary and General Comments**

Reviewer #1: The authors have completed the requested information and the manuscript is suitable for publication.

Reviewer #3: All the points raised in the first review were adequately addressed in this new version of the manuscript, and no further modifications are necessary.

Reviewer #4: This is a important and well-designed study. The main finding was the intracellular location of the Tc24 protein and its high expression in the flagellum of trypomastigotes and reduced in replicating amastigotes. Minor modifications are necessary, but these do not interfere with the quality of the study. The binomial nomenclature needs to be revised throughout the text, such as "T. Congolese" in line 231, T. cruzi not italicized in line 338 and 386, among others. The hybridoma clone number needs to be corrected in line 294. In line 398, T. cruzi amastigotes are transforming back to trypomastigotes, not to amastigotes. Finally, in line 501 the 28H post infection should actually be 48H.

PLOS authors have the option to publish the peer review history of their article (what does this mean?). If published, this will include your full peer review and any attached files.

Reviewer #1: No

Reviewer #3: No

Reviewer #4: No

---

## [Editor Report · Acceptance letter]

18 Aug 2021

Dear Dr. Pollet,

We are delighted to inform you that your manuscript, "Location and expression kinetics of Tc24 in different life stages of Trypanosoma cruzi," has been formally accepted for publication in PLOS Neglected Tropical Diseases.

Best regards,

Shaden Kamhawi

co-Editor-in-Chief

Paul Brindley

co-Editor-in-Chief
